# Rapid re-identification of human samples using portable DNA sequencing

**Sophie Zaaijer[1,2†]\*, Assaf Gordon[2], Daniel Speyer[1,2], Robert Piccone[3], Simon Cornelis Groen[4], Yaniv Erlich[1,2,5]\***

[1]Department of Computer Science, New York Genome Center, New York, United States; [2]New York Genome Center, New York, United States; [3]Data Science Institute, Columbia University, New York, United States; [4]Department of Biology, Center for Genomics and Systems Biology, New York University , New York, United States; [5]Department of Computer Science, Fu Foundation School of Engineering, Columbia University, New York, United States

**Abstract** DNA re-identification is used for a broad suite of applications, ranging from cell line authentication to forensics. However, current re-identification schemes suffer from high latency and limited access. Here, we describe a rapid, inexpensive, and portable strategy to robustly re-identify human DNA called 'MinION sketching'. MinION sketching requires as few as 3 min of sequencing and 60-300 random SNPs to re-identify a sample enabling near real-time applications of DNA re-identification. Our method capitalizes on the rapidly growing availability of genomic reference data for cell lines, tissues in biobanks, and individuals. This empowers the application of MinION sketching in research and clinical settings for periodic cell line and tissue authentication. Importantly, our method enables considerably faster and more robust cell line authentication relative to current practices and could help to minimize the amount of irreproducible research caused by mix-ups and contamination in human cell and tissue cultures.
DOI: https://doi.org/10.7554/eLife.27798.001

**\*For correspondence:**
sophie@cornell.edu (SZ);
yaniv@cs.columbia.edu (YE)

**Present address:** †Jacobs Technion-Cornell Institute at Cornell Tech, New York, United States

DNA is a powerful biometric identifier. With the exception of monozygotic twins, DNA profiles are unique to each individual on Earth (*Kayser and de Knijff, 2011*; *Bieber et al., 2006*; *Gymrek et al., 2013*). The ability to re-identify DNA has multiple applications in a broad range of disciplines. In research settings, re-identification is employed to authenticate cell lines and patient-derived xenografts (PDXs) by matching their DNA to validated genomic profiles (*NIH, 2016*; *AMS, 2015*; *El-Hoss et al., 2016*). In clinical genetics, the American College of Medical Genetics recommends using DNA genotyping tests to track sample identity and avoid sample mix-ups during clinical whole genome/exome sequencing (*Green, 2013*). In forensics, DNA re-identification has become one of the most common techniques to identify samples from crime scenes or from victims of mass disasters, and human trafficking (*Kayser and de Knijff, 2011*). Despite this wide range of applications, current DNA re-identification methods suffer from high latency and lack of rapid access.

Cell line contamination is a widespread and persistent problem in academic, clinical and commercial research, despite having been recognized for 50 years (*Gartler, 1968*; *Alston-Roberts et al., 2010*; *Yu et al., 2015*; *Almeida et al., 2016*). The ongoing publication of irreproducible research is a major economic burden on society and costs 28 billion dollars each year in the USA alone (*Capes-Davis and ICLAC, 2016*). To diminish this the NIH and various journals require researchers to authenticate cell lines by matching their DNA profiles to validated signatures (*NIH, 2016*; *AMS, 2015*). The most common DNA re-identification strategy is to genotype a minimum of eight autosomal polymorphic short tandem repeats (STRs) (*Masters et al., 2001*; *Smith et al., 2012*; *Capes-Davis et al., 2010*; *Reid et al., 2013*; *ATCC, 2011*). However, this technique entails the use

**eLife digest** The human genome represents the complete set of genetic information needed to make a person. DNA sequencing technologies used to study genomes have become much faster, cheaper and more accessible over recent years. This has enabled them to be used more regularly in various fields like precision medicine, in research laboratories and forensics. Even so, there are still fields where optimization is critical.

Reproducibility is an important issue in biomedical research; one group of scientists working with human cells may report results that other scientists cannot reproduce. Sometimes this is because the original work was done in the wrong type of cells by mistake. Human cells used in biomedical research are very hard to discriminate from each other using microscopes; however, DNA analysis can be used to ensure the origin of the cells.

The MinION device, a USB compatible handheld DNA sequencer, has become available in the last few years. Its size, speed and portability could enable many new uses for DNA sequencing. Technology like this could be used to confirm which cells the scientists are working with before they publish their results. Yet, currently DNA readings from the MinION are not accurate enough to be used to reliably confirm the identity of human cells used in research.

Zaaijer et al. have now developed an approach that can accurately identify human cells using the MinION device. The approach involves "DNA re-identification", which works by comparing an unknown DNA sample to a collection of known DNA profiles. Using their new method, Zaaijer et al. report that, with three minutes of DNA sequencing, they can correctly identify a DNA sample, with 99.9% confidence. This is a high enough level of accuracy for the system to tell the difference between one person and another, using only their DNA.

This new technology is much faster than current rapid DNA sequencing approaches. Previously, processing DNA samples could take hours or even days and was not particularly portable. The new technology has many applications from finding criminals to diagnosing illnesses and tracking epidemics. It is also an affordable way for laboratories to confirm the identity of cells they are working with. This has the potential to save billions in research funding each year and speed up scientific progress.

DOI: https://doi.org/10.7554/eLife.27798.002

of time consuming PCR-based steps and specialized capillary electrophoresis machines, with the latter not being part of standard laboratory equipment.

The STR profiling method for cell line authentication originates from the forensic sciences, where it is standard practice (*Alston-Roberts et al., 2010*; *Almeida et al., 2016*). Since most cell lines are uniquely derived from single patients, the STR profiling method was adopted for cell line authentication as well. However, the unstable genetic nature of cancer cell lines undermines the usefulness of these standards and affects the re-identification efficiency of the multi-allelic STR markers (*Capes-Davis et al., 2010*; *Alston-Roberts et al., 2010*; *Castro et al., 2013*). Previous studies have explored using more stable SNP markers for re-identification. Indeed, a carefully selected panel of ~50 SNPs confers a power for re-identification similar to that provided by the 8–13 STR markers used in forensics and cell line authentication (*Jobling and Gill, 2004*; *Sanchez et al., 2006*; *Yu et al., 2015*). SNPs are increasingly being used for cancer cell line authentication (*Sanchez et al., 2006*; *Castro et al., 2013*; *Otto et al., 2017*).

To overcome current latency and accessibility issues, we have developed a rapid and novel SNP-based strategy for robust re-identification of human DNA using a MinION sequencer (produced by Oxford Nanopore Technologies, ONT). The MinION is a cheap and portable DNA sequencer that weighs only 100 g and can be plugged into a laptop computer. This device can easily be adopted as part of standard laboratory equipment. Our SNP-based strategy, termed 'MinION sketching', exploits real-time data generation by sequentially analyzing extremely low coverage shotgun-sequencing data from a sample of interest and comparing observed variants to a reference database of common SNPs (*Figure 1*). We specifically sought a strategy that does not require PCR in order to maximize speed, reduce the number of steps in the protocol and to omit species bias that follows from using human-specific primers (*Alston-Roberts et al., 2010*). However, this poses two technical

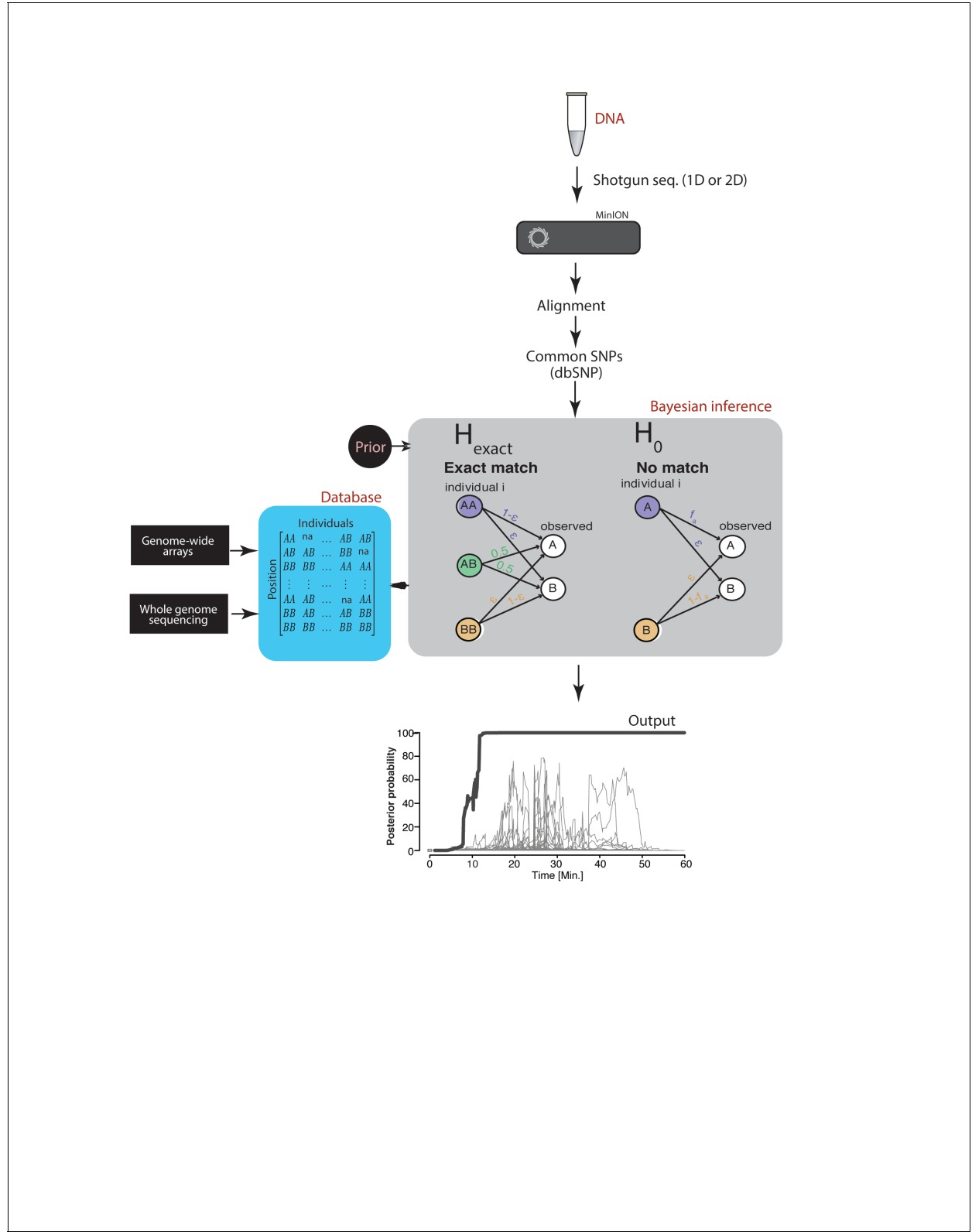

**Figure 1.** Schematic overview of MinION sketching. A DNA sample is prepared for shotgun sequencing. Libraries are prepared either for 1D or 2D MinION sequencing (without and with hairpin, respectively). Variants observed in aligned MinION reads are only selected if they coincide with known polymorphic loci while others are treated as errors. These SNPs are compared to a candidate reference database comprised of samples genotyped with whole genome sequencing or sparse genome-wide arrays (~600K-900K SNPs per candidate file). A Bayesian framework computes the posterior

*Figure 1 continued on next page*

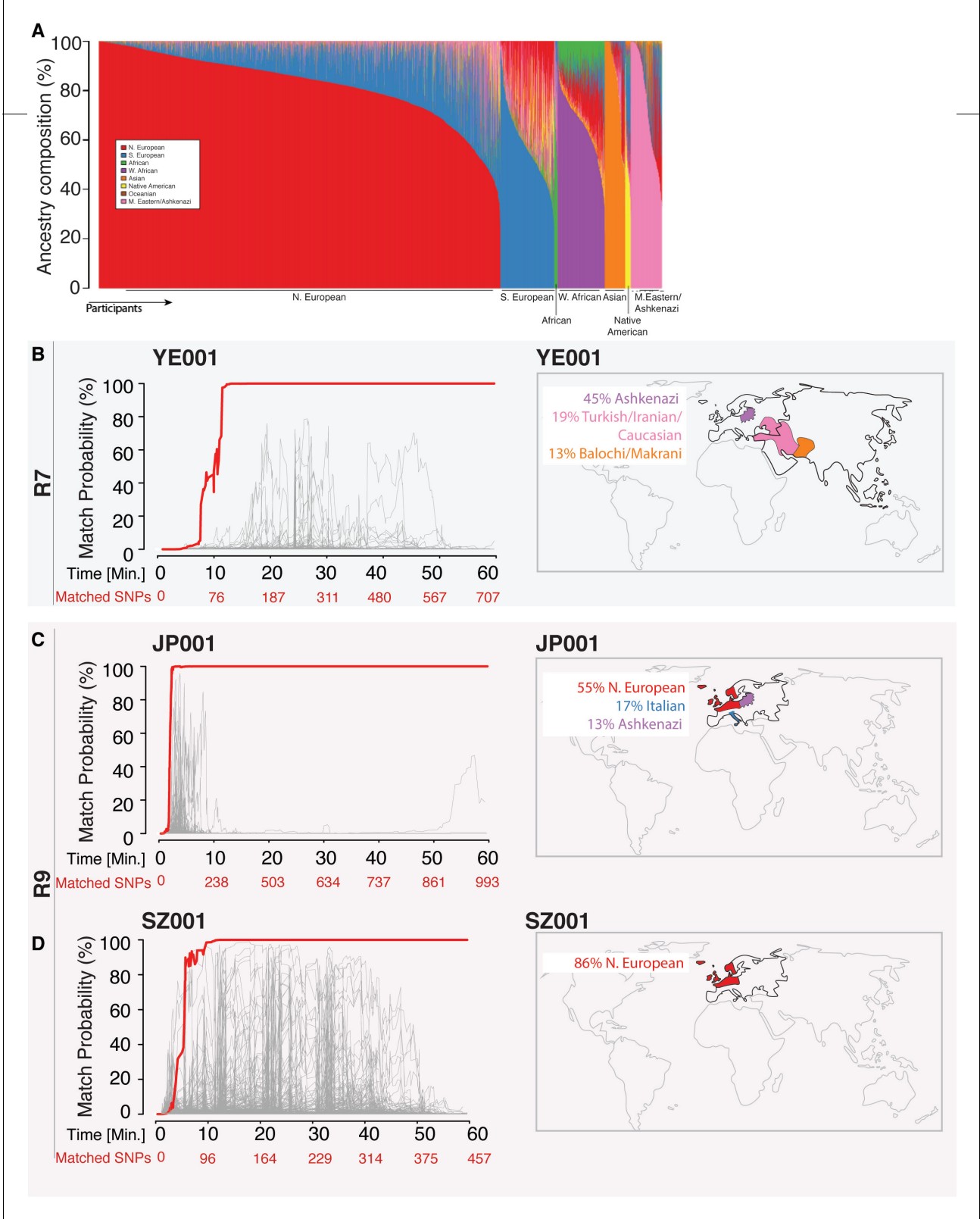

**Figure 2.** Re-identification of three DNA samples against a database with 31,000 individuals. (**A**) A Frappe plot showing the population structure of the database with a collection of 31,000 genome-wide SNP arrays. (**B–D**) The match probability is inferred by comparing a MinION sketch to its reference file as a function of the MinION sketching time (red line) and the number of SNPs analyzed. The prior probability for a match was set to $10^{-5}$. The match probabilities are inferred by comparing the MinION sketches to a database with 31,000 genome-wide SNP arrays (including the matched

*Figure 2 continued*

individuals). Right: Ancestral background of the corresponding individuals; only ancestry predictions of >10% are indicated. (B) The DNA sample was collected from an Ashkenazi-Uzbeki male (YE001) and sequenced using R7 chemistry. (C) The sample was collected from a Northern European female (SZ001) and sequenced using R9 chemistry. (D) The sample was collected from a Northern European-Italian-Ashkenazi male (JP001) and sequenced using R9 chemistry.

DOI: https://doi.org/10.7554/eLife.27798.004

The following figure supplement is available for figure 2:

**Figure supplement 1.** A prior representing a database larger than the world population still allows for identification power.

DOI: https://doi.org/10.7554/eLife.27798.005

challenges. First, MinION sequencing exhibits a high error rate of 5–15% (*Ip et al., 2015*), which is two orders of magnitude beyond the expected differences between any two individuals. Second, MinION sketching produces shotgun-sequencing data that only covers a fraction of the human genome due to the limited capacity of a MinION cell. As such, the extremely low coverage dictates that each locus is covered by up to one sequence read, which nullifies the ability to enhance the signal by integrating multiple reads or observing both alleles at heterozygous loci. Taken together, these challenges translate to a noisy identification task where the available genotype data only provide a mere sketch of the actual genomic data.

To address these challenges, we developed a Bayesian algorithm that computes a posterior probability that the sketch matches an entry in the reference database ($H_{exact}$) or has no match in the database, taking into account each marker's allele frequency and the prior probability that a sample matches an entry in the reference database. The Bayesian approach sequentially updates the posterior probability with every new marker that is observed until a match is found. Collectively, our method can robustly re-identify a sample without PCR amplification, yet with very high probability, overcoming the low coverage and high error rate from nanopore sequencing.

## Results

In order to benchmark our re-identification method for real-life applications, we tested it in a variety of technical scenarios. To start, we constructed two large-scale proof-of-principle reference databases of genomic datasets that would stress the specificity of our method. The first reference database contains 31,000 genome-wide ~600K-900K SNP genotyping array files from individuals tested by Direct-to-Consumer (DTC) companies such as 23andMe, AncestryDNA, and FamilyTreeDNA

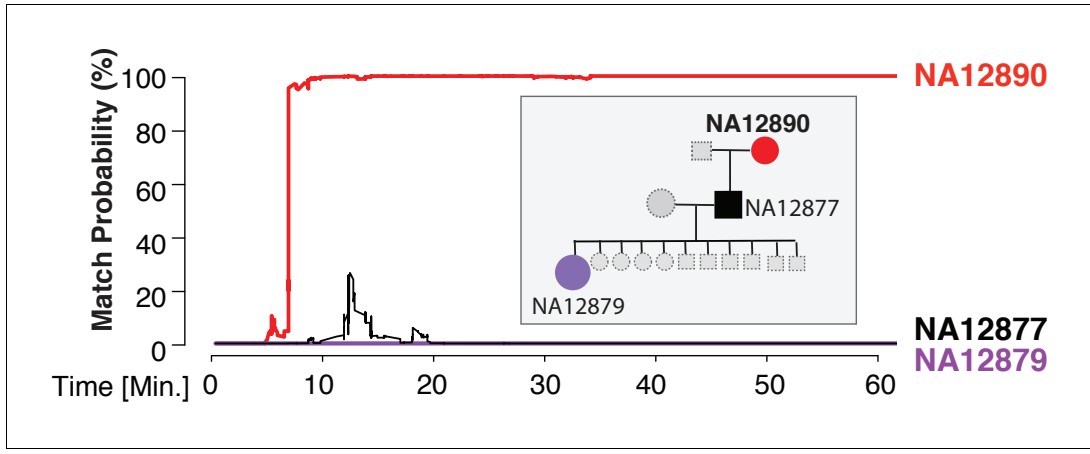

**Figure 3.** Re-identification of HapMap sample NA12890. The match probability is inferred by comparing a MinION sketch of NA12890 to the reference files of her own genome (red), her son's genome (black), and her granddaughter's genome (purple), as a function of the MinION sketching time (red line). The prior probability for a match was set to $10^{-5}$. Inset: the pedigree of 1000Genomes sample NA12890

DOI: https://doi.org/10.7554/eLife.27798.007

(*Figure 2A*) (*Erlich, 2015*). The second reference database consists of genome-wide ~700K-800K SNP genotyping array files from 1,099 cancer cell lines in the Cancer Cell Line Encyclopedia (CCLE) (*Barretina et al., 2012*) and can be used for cell line authentication. Next, we created a MinION sketch for DNA samples in multiple technical scenarios (*Supplementary file 1A*). These testing scenarios included extracting DNA from a spit kit or cell line culture, sequencing with either the R7 chemistry or the newer R9 chemistry, and re-identifying a sample with a genetically unstable background. The genetic reference files for each of these sketched samples were included in our reference databases.

We found that the MinION sketching procedure re-identified human DNA with high accuracy after just minutes of operation. After 13 min of sketching using the R7 chemistry, the Bayesian algorithm re-identified the NA12890 sample (a female CEU individual from the HapMap project) with a posterior probability greater than 99.9%. Despite the high error rate of this relatively old chemistry and the low coverage, the algorithm needed only 195 bi-allelic variants to re-identify the sample (*Figure 3* and *Supplementary file 1B*). This is only ~2 times above the theoretical expectation for re-identifying a person by fingerprinting random markers (*Lin et al., 2004*). To further test the robustness of our method, we re-sketched NA12890's sequencing data against reference files for her first-degree relative (NA12877) and second-degree relative (NA12879). Importantly, no exact-matching probability was observed, highlighting the specificity of our method (*Figure 3*).

Using the 31,000-individual reference database (consisting of genetic profiles from individuals genotyped by DTC companies), we repeated the R7 experiment with a sample of a mixed Ashkenazi-Uzbeki male (YE001). Again, we were able to re-identify this person within 13 min after assessing 110 SNPs (*Figure 2B* and *Supplementary file 1B*), further showing that the method produces consistent results across ethnic origins. None of the other 31,000 individuals reached this level of matching probability (*Figure 2B*). Finally, given that the number of reference samples in our database is in the thousands, but the number of people in the world in the billions, we wondered about the impact of the prior probability on identifying individuals. To this end, we tested various prior probabilities of identifying the YE001 sketch. We found that the initial selection of the prior probability had no effect on the matching ability and only slightly increased the time required to achieve a high-confidence match. Even with a prior probability that considers a database around a million times bigger than the world's population ($10^{15}$), the posterior probability reached 99.9% with only 25 min of sketching YE001 (*Figure 2—figure supplement 1*), showing that our method returns robust results regardless of the chosen prior.

Moving to the newer R9 chemistry provided even faster re-identification results. We sketched samples of a Northern European female (SZ001) and a Northern European-Italian-Ashkenazi male (JP001) using this chemistry. We were able to re-identify these two samples using only 98–134 SNPs, and the fastest identification required fewer than 5 min of MinION sketching (*Figure 2C and D* and *Supplementary file 1C*). Again, none of the other 31,000 individuals in our database were matched to SZ001 or JP001 using this strategy. The rapid re-identification seems to be linked intimately to the increased speed with which DNA strands pass through the pores with the R9 chemistry versus the R7 chemistry (250bases/sec *vs* 70bases/sec). These results suggest that future developments in speeding up the DNA reading time could further reduce the re-identification time.

Next, we explored the applicability of MinION sketching for cancer cell line authentication, a longstanding issue in the research community. We used MinION sketching and the R9 chemistry to authenticate THP1, a monocytic leukemia strain, against the second reference database that consisted of cell lines from the CCLE. To show that more than one sample can be authenticated at the same time, we barcoded the THP1 sample and combined it with an additional, barcoded human sample. From the barcoded THP1 reads that were generated in ~3 min of sequencing, the sketching procedure leveraged 91 SNPs to authenticate the THP1 cell line with a posterior probability of 99.9%. None of the other 1098 CCLE reference files reached a probability of 99.9% or even exceeded a 10% match probability (*Figure 4A*, *Supplementary file 1D*).

Thus far, re-identification required an intersection of 91–195 SNPs from the MinION sketch and reference SNP file to reach a match probability of 99.9%. Having observed this range in the number of SNPs required, we wished to find the minimum number of intersected SNPs necessary to obtain a 99.9% match. This way we can optimize the sequencing time. To determine such a 'stop sketching' threshold, we simulated 10,000 different sketching runs for the THP1 cell line (*Figure 4B*) and SZ001 (*Figure 4—figure supplement 1A*). The majority of simulated MinION sketches reached a match

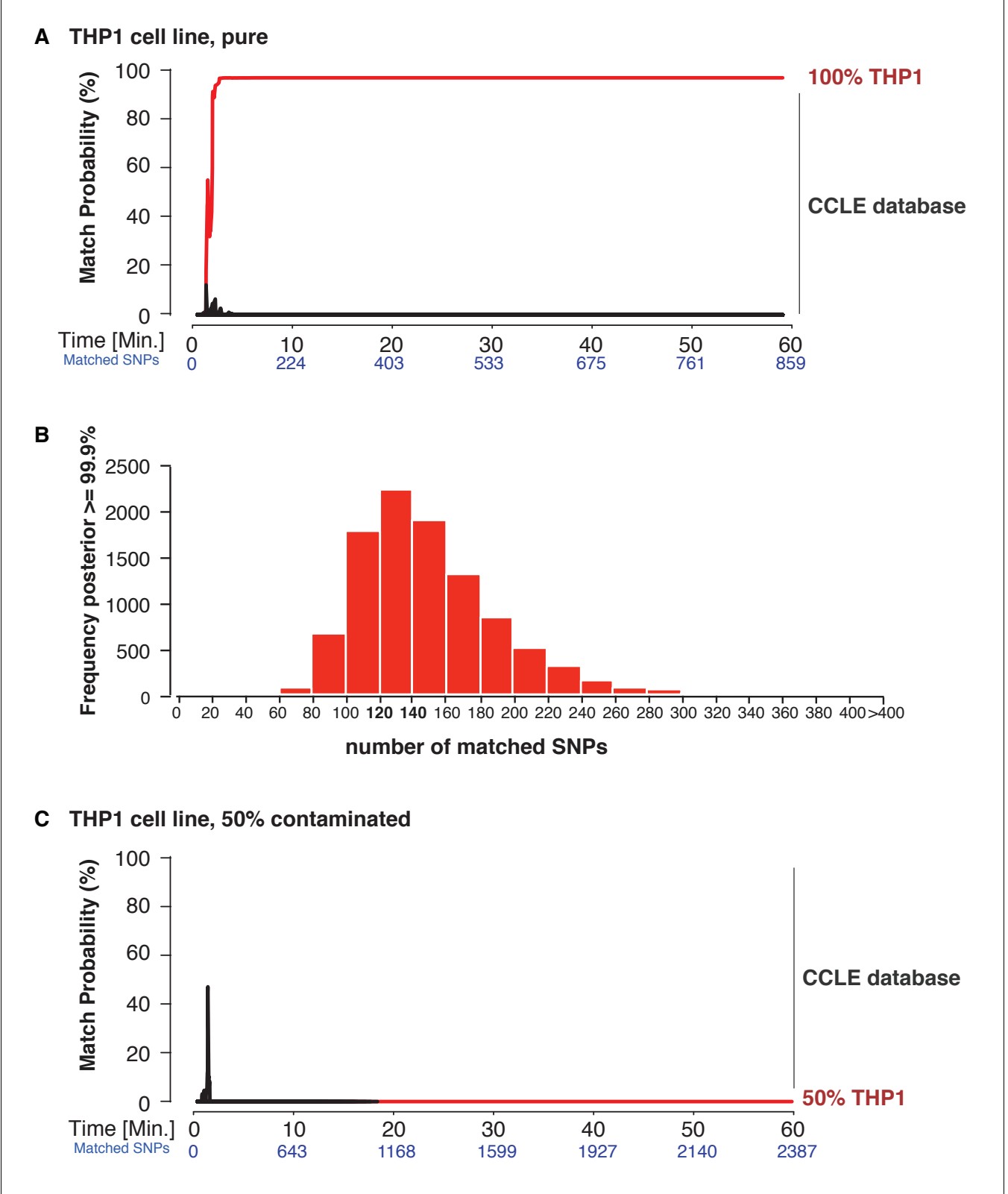

**Figure 4.** Cell line authentication. Barcoded DNA from the THP1 cell line is mixed 1:1 with a random, barcoded sample. Analysis of only the THP1 reads was used to infer 'pure' matches, while analyses of the mixture were used to characterize the efficiency of matching using contaminated samples. The match probability is inferred by comparing a MinION sketch to 1,099 reference files that are part of the cancer cell line encyclopedia (CCLE) generated by the Broad Institute (grey). (**A**) The posterior probability for an exact match between the MinION sketch of the 'pure' cell line THP1

*Figure 4 continued on next page*

*Figure 4 continued*

(considering a single barcode) and the reference file generated by the CCLE (the red line indicates the THP1 reference file, other strains are depicted in grey). The posterior probability is plotted as a function of the sketching time and number of SNPs analyzed. (**B**) 10,000 simulated runs of sketching the THP1 cell line were matched against its reference file. The number of SNPs used to reach a 99.9% match (x-axis), is plotted against the number of times it is observed (y-axis). (**C**) The posterior probability that the contaminated (50% mixed) sample matched THP1 is plotted as a function of the sketching time and number of SNPs analyzed.

DOI: https://doi.org/10.7554/eLife.27798.008

The following figure supplement is available for figure 4:

**Figure supplement 1.** Cell line authentication.

DOI: https://doi.org/10.7554/eLife.27798.009

with a 99.9% probability using only 120–140 intersected SNPs. By 300 intersected SNPs, 99.6% of all sketches of the THP1 cell line were matched to its reference file with a probability of 99.9%, and for sketches of SZ001 by 240 SNPs that intersected with its reference. As expected, none of our simulation files failed to reach a correct match with a 99.9% probability with the correct reference file in the database. Although the number of mismatches per run was strongly correlated with the number of SNPs analyzed (*Figure 4—figure supplement 1B*), the results from our sequencing runs and simulations suggest that even genetically unstable cancer cell lines can be identified with confidence using no more than 300 SNPs. The minimum sequencing run time necessary to infer a match depends on the yield of the specific run and the chemistry used. In summary, the MinION sketching method relies on the presence of the reference file in the database. If computing the posterior probability for 300 SNPs does not result in a 99.9% match, then the reference file for that cell line or individual is almost certainly not present in the reference database and further sketching is highly unlikely to yield any success.

Next, we wondered how a severe contamination with cells of another origin would affect successful cell line authentication. Cell line cross-contamination is caused mostly by overgrowth from secondary cell lines with a substantially shorter generation time (*Capes-Davis et al., 2010*; *Alston-Roberts et al., 2010*). To start assessing the effects of contamination, we re-analyzed the data from the THP1 experiment but without resolving the barcodes, which essentially reflects a 50% contamination. The algorithm correctly showed a 0% match probability to the THP1 reference file or any other cell line in the database (*Figure 4C*). We further explored the effect of the fraction of contamination on matching sketches with the THP1 reference file. By sampling from the above data in different proportions, we found that the algorithm correctly rejects a match for samples with contamination levels above 25% (*Figure 5*). While it may seem that the algorithm is not as sensitive to contamination as current STR-based methods, periodic testing of a cell culture with our method will reveal the contamination in a more timely fashion (see Discussion).

Lastly, we aimed to explore a sample preparation strategy that requires minimal hands-on time. To this end, we utilized a simple protocol to extract DNA using the rapid transposase-mediated fragmentation and adaptor ligation kit provided by ONT. This method generates 1D reads, where only one of the two strands passes through the nanopore, resulting in reads with a higher error rate (*Supplementary file 1E*). The advantage of this method is the speed and convenience of the preparation protocol. In only 55 min, we were able to extract DNA and produce a ready-to-sequence library (*Figure 6A*, example of execution: *Figure 6—video 1*). The increased error rate resulted in the requirement for more SNPs to reach the re-identification threshold. In our experiment with the rapid sample preparation protocol we needed 239 SNPs to identify SZ001 with >99.9% probability (*Figure 6B*). As such, re-identification of DNA and cell line authentication can still be completed with the same level of accuracy in one afternoon and using only minimal hands-on time by the researcher.

## Discussion

Our results show that MinION sketching for re-identification of human samples is robust and is faster than other currently available methodologies. With its high accuracy, it can be adopted for the periodic testing and authentication of cell lines in research settings, for verifying biobank entries, for tracing samples in clinical genetics, and for certain forensic applications. Based on only 3–13 min of

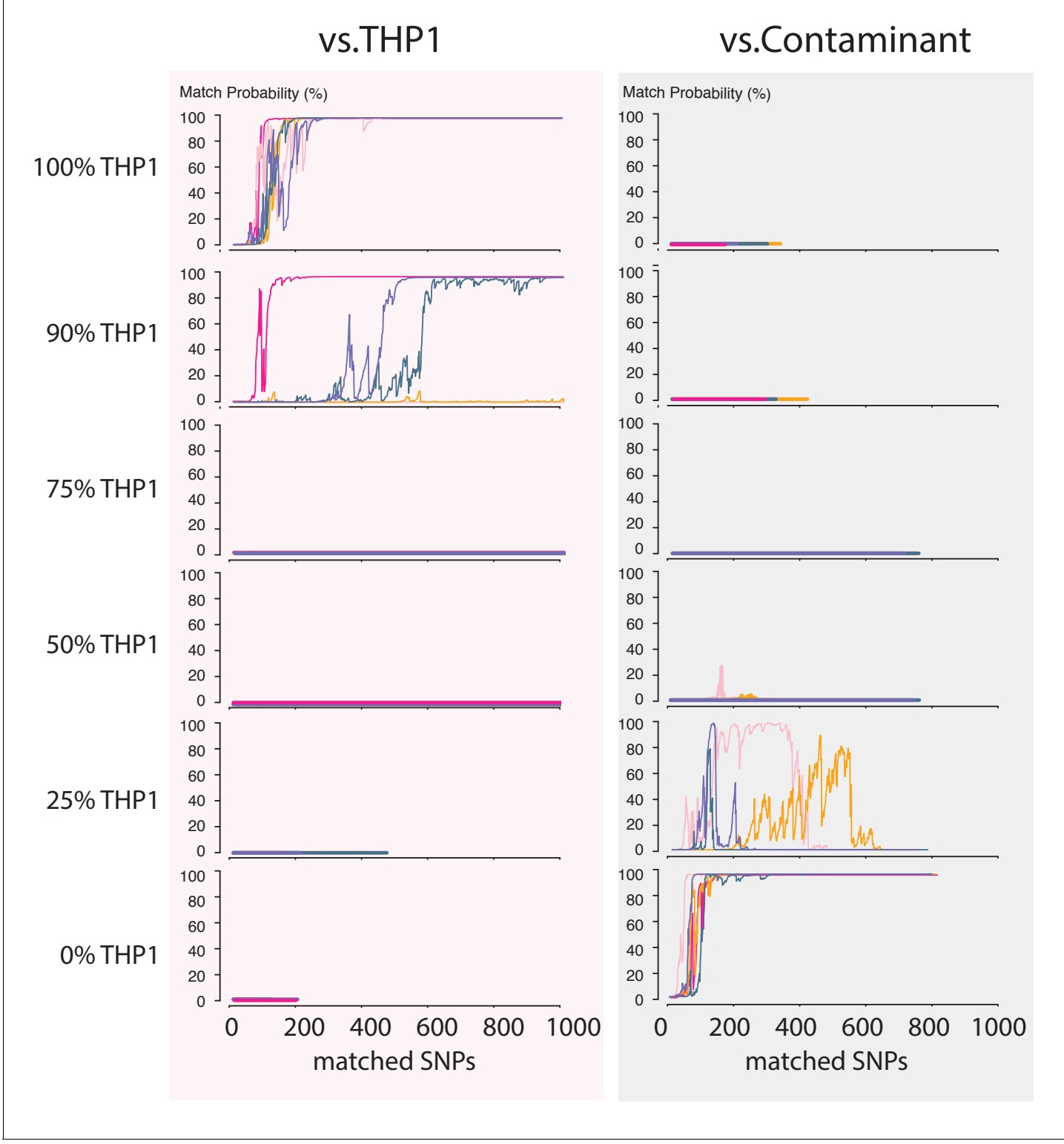

**Figure 5.** Contamination simulations. Random reads from a run with DNA from THP1 cells and a random, barcoded sample (the contaminant) are mixed in the indicated proportions and shuffled. This simulated MinION sketch is matched against the THP1 reference file, and the contaminant reference file. This process is repeated five times for each simulated contamination (pink, light-pink, purple, green and yellow lines). The match probability here is a function of the number of SNPs analyzed.

DOI: https://doi.org/10.7554/eLife.27798.010

The following figure supplement is available for figure 5:

*Figure 5 continued on next page*

*Figure 5 continued*

**Figure supplement 1.** Theoretical effect of differences in doubling time of contaminants in a cell culture.

DOI: https://doi.org/10.7554/eLife.27798.011

sequencing and 60–300 informative SNPs, MinION sketching can infer the identity of an anonymous sample. It is a unique addition to current state-of-the-art DNA re-identification methodologies.

## Rapid on-site detection of sample mix-up and contamination

The main cause of cell line mix-ups is suggested to be human error (*Alston-Roberts et al., 2010*; *Yu et al., 2015*; *Almeida et al., 2016*). It is therefore crucial to have means to monitor these errors rapidly and periodically. While the American Type Culture Collection (ATCC) offers an STR-based cell line authentication service, the overall procedure requires shipping consumables and samples back-and-forth and takes 2 weeks to complete. This works sufficiently in situations of cell line contamination that originate from mislabeling of a cell culture (100% contamination). Yet, a processing time of 2 weeks is suboptimal when caused by the mistaken transfer of cells from one culture to another, which can lead to cases of fitness competition between the cell lines (*Alston-Roberts et al., 2010*; *Yu et al., 2015*). It takes only 10 cells from a line with a doubling time that is 2–4 hr shorter than that of the original strain to overgrow an initial culture ($10^6$ cells) within 2 weeks (*Figure 5—figure supplement 1*), which would currently be the time-point when STR typing results would be returned by the ATCC.

Strikingly, once cells are in log-phase it can take as few as 2 days to change the contamination level of a culture from 1% to 80%. Our contamination simulations show that a contamination $\geq 25\%$, and often less than that, precludes a true matching result. Although the current STR-based methods can pick up on lower levels of contamination, in practice this does not make much difference considering: (1) the pace with which a contaminant can invade a cell culture, and (2) the relatively low identification speed of methods currently employed that precludes the timely return of data on the genomic composition of a cell population over multiple time points. Moreover, STR analysis is typically done using human-specific primers for amplification, and this therefore limits the identification of contaminants to ones of human origin (*Alston-Roberts et al., 2010*). Our method, on the other hand, does have the potential ability to detect DNA from contaminants of non-human origin, such as

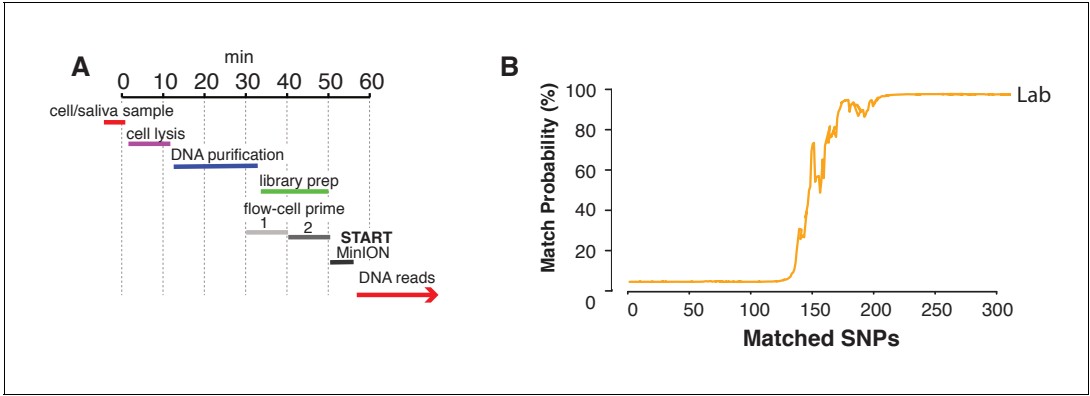

**Figure 6.** Rapid library preparation. (A) Schematic of the steps from sample to MinION sketch. The current method requires ~55 min until the MinION starts to generate reads. (B) The match probability is inferred by comparing a MinION sketch generated by transposase-mediated adaptor ligation (the rapid kit) to its reference file as a function of the number of SNPs analyzed. The prior probability for a match was set to $10^{-5}$. The rapid library protocol was tested in the lab. The MinION sketch was generated from sample SZ001. The library was prepared in 55 min in the laboratory. After analyzing 239 informative SNPs the posterior match probability exceeded 99.9%.

DOI: https://doi.org/10.7554/eLife.27798.012

The following video is available for figure 6:

**Figure 6—video 1.** The movie depicts the rapid, on-site library preparation protocol using the Bento Lab (www.bento.bio) for DNA extraction and library preparation, prior to starting DNA sequencing as described in *Figure 6*.

DOI: https://doi.org/10.7554/eLife.27798.013

infectious organisms like mycoplasma. Such contaminants could be identified efficiently when our pipeline is run in combination with metagenomic methods for real-time microbial detection (as in *Quick et al., 2015*). The key to detect cell line contamination with human and non-human cells is periodic testing.

The MinION can be part of standard lab equipment and facilitates rapid sample preparation and testing just prior to key experiments. We show that with our MinION sketching method, cell line authentication can be achieved in the lab in one afternoon, either using a hands-on or hands-off protocol. The first protocol involves a hands-on ~3 hr library preparation step (including DNA extraction), but after only ~3 mins of sequencing we were able to identify the THP1 cell line out of 1,099 cancer cell lines with a posterior probability of 99.9%. The second protocol requires just 55 min for DNA extraction and transposase-mediated adapter ligation, after which sequencing can start. Our MinION sketching method reduces re-identification latency so that research does not have to be paused for long until the DNA profiling results return.

## MinION sketching is neither affected by marker dropout nor by a genetically unstable genome

Our method relies on randomly sampling SNPs from the genome, instead of a fixed set of small numbers of STRs or SNPs in a panel. This way we can omit a time consuming and biased PCR step in our method, and avoid the loss of statistical power caused by allelic dropout. This is particularly advantageous for cancer cell line authentication where genomic instability is prevalent. Cancerous cell lines commonly undergo loss of heterozygosity or exhibit aneuploidy, which affect STR-based re-identification of DNA samples through the loss of alleles (*Capes-Davis, 2013*). Furthermore, cancer cells that are deficient in their mismatch repair (MMR) pathways and suffer from microsatellite instability are identified more accurately by SNP-based than by STR-based identification methods (*Castro et al., 2013*; *Otto et al., 2017*). Because of these challenges, the current official ASN-0002 standards for STR analysis use an 80% matching threshold to positively match the STR profile in question to a reference file. Using this threshold, cell lines can be identified correctly in 98% of the cases (*Capes-Davis, 2013*). In our experiments, we see a clear correlation between the number of mismatches and the SNPs that are used to infer a match. The occurrence of mismatches in matching the DNA from a genetically unstable cancer cell line to its reference file results in a need to collect more SNP evidence for a match. Still, when we simulated matching the THP1 cell line to its reference file 10,000 times we found that intersecting a minimum of only 300 SNPs leads to a correct match in 99.6% of the cases. Furthermore, we found that based on this simulation, the SZ001 simulation and our experimental data, 300 SNPs could be used as a 'stop-sketching' threshold. Importantly, we never observed a false-positive match to any other reference file in the CCLE database. As a SNP-based method MinION sketching improves the precision of re-identifying cancer cell lines compared to the STR-based identification methods. ANSI-approved standards for SNP usage in cancer cell line authentication would be useful to promote the community-wide adoption of SNP-based sample re-identification (also proposed by *Yu et al., 2015* and *Otto et al., 2017*, among others).

## Costs of re-identification

The start-up cost for the MinION is currently $1000, and multiplexing 12 DNA samples in one run makes the cost of consumables for sequencing a sample around $100. This cost per sample is already lower than the ATCC STR-typing service, which is $195, but higher on a per-run basis than the Geneprint system method. However, the latter method can only be used with access to the Applied Biosystems 3500 platform, and involves a more elaborate protocol that requires hands-on time, therefore incurring higher costs of labor. Although the balance between cost of labor and costs of machine depreciation and consumables poses a trade-off for all methodologies, the requirement of extensive hands-on laboratory work seems a main driver for avoiding authentication with current STR-based tests. Given that MinION sketching requires only minimal hands-on time and provides re-identification within hours instead of days/weeks, it is a very efficient and competitive re-identification method, especially when working with a small number of samples. While the costs of MinION sequencing continue to decrease, MinION sketching is currently not competitive in price for high-throughput testing until sequencing costs will have decreased further.

## Concluding remarks

In conclusion, to help solve the long-standing issue of (cancer) cell line contamination and to enhance the traceability of tissue samples in biobanks we developed an rapid re-identification method for DNA samples. Our method lowers the barrier for adoption of regular cell line authentication, which is important since only periodic testing will detect contamination and mix-ups efficiently and reduce the costs involved with irreproducible research. MinION sketching can easily be done in laboratories, in the clinic, or in biobanks as a routine sample authentication test.

# Materials and methods

## The Bayesian matching algorithm

The matching algorithm uses a Bayesian framework to evaluate the posterior probability of a match. Let $x_i \in \{Y, N\}$ be a random variable that either indicates whether the MinION sketch directly matches a known person ($x_i = Y$), or does not match ($x_i = N$) with respect to the $i$-th individual in the database. Let $D_k$ be the observed MinION data for the $k$-th bi-allelic marker, with $D_k \in \{A, B\}$, where $A$ and $B$ denote the two alleles; and let $D = (D_1, D_2, \dots, D_n)$ denote the observation for $n$ bi-allelic markers.

The posterior probability of the matching outcome for the $i$-th sample is:

$$p(x_i|D) = \frac{p(x_i) \cdot p(D|x_i)}{p(D)} \tag{1}$$

where $p(x_i)$ is the prior probability for the matching status of $i$-th sample and is specified by the user.

The likelihood is approximated using the following equation:

$$p(D|x_i) = \prod_{kn\{1,\dots,n\}} p(D_k|x_i) \tag{2}$$

The likelihood of an exact match given the data of the $k$-th marker, $p(D_k|x_i = Y)$, is given by the following matrix:

$$M = \begin{array}{c} \\ \begin{bmatrix} 1-\epsilon & \epsilon \\ 0.5 & 0.5 \\ \epsilon & 1-\epsilon \end{bmatrix} \end{array} \begin{array}{c} A \quad B \\ \\ AA \\ AB \\ BB \end{array} \tag{3}$$

where the rows denote the genotype of the $i$-th sample for the $k$-th marker as observed in the DNA database, the columns correspond to the observed genotype in the MinION data, and $\in$ denotes the error rate assuming symmetry in confusing allele $A$ for allele $B$ and vice versa. $p(D_k|x_i = Y)$ corresponds to a specific row of **M** based on the observed genotype of a sample in the database. For example, if the genotype of the database sample is $AA$, then $p(D_k = A|x_i = Y) = 1 - \epsilon$ and $p(D_k = B|x_i = Y) = \epsilon$.

The likelihood of a *mismatch* given the data of the $k$-th marker, $p(D_k|x_i = N)$, basically corresponds to observing the allele $D_k$ in a random person from the population. This probability is the sum of two processes: (i) the random person has the same allele as $D_k$ and the observation is error-less or (ii) the random person does not have the same allele as $D_k$ but a sequencing error flipped the observed allele. Therefore:

$$p(D_k|x_i = N) = (1 - \epsilon) \cdot f(D_k) + \epsilon \cdot [1 - f(D_k)] \tag{4}$$

where $f(D_k)$ denotes the frequency of the observed allele in the population

Finally, the evidence, $p(D)$ is given by:

$$p(D) = \sum_{x_i \in \{Y,N\}} p(x_i) \cdot p(D|x_i) \tag{5}$$

## DNA samples for sequencing

We purchased the genomic DNA sample for the 1,000 Genomes individual NA12890 from the Coriell Institute. The THP1 cell line (ECACC Cat# 88081201, RRID:CVCL_0006) was used from laboratory resources and its authenticity was thoroughly verified in this study. YE001 and SZ001 were derived from the corresponding authors (Y.E. and S.Z.) using cheek swabs (Catch-All Sample Collection Swab Epicentre QEC89100) or a saliva collection kit (*Supplementary file 1A*). JP001 was sampled through saliva collection.

## DNA preparation for 2D sequencing

Genomic DNA from NA12890 and YE001 (*Supplementary file 1A*; exp. 1, exp. 2 respectively) were prepared for 2D MinION libraries using the SQK-MAP006 kit (ONT) as described by *Zaaijer et al. (2016)*. 2D libraries are double-stranded DNA fragments with a ligated hairpin loop and adaptors containing a tether and motor protein necessary for MinION sequencing, and were run on the R7 flow cells. DNA samples from SZ001, JP001 and the THP1 cell line were prepared using the SQK-NSK007 kit from ONT (*Supplementary file 1A*; exp. 3, exp. 4, exp. 5) and run on R9 flow cells.

## Rapid library preparation in the lab

Samples (*Supplementary file 1A*, exp. 6) were collected by cheek swab (Catch-All Sample Collection Swab Epicentre QEC89100) scraping ~30 s on both sides of the cheek. Cells were recovered in 200 µl PBS. After addition of 20 µl Proteinase K and 200 µl lysis buffer (DNeasy blood and tissue kit, Qiagen, #69504), the sample was incubated at 56°C for 10 min. The sample was then applied to the column, spun for 1 min, and washed sequentially with buffers AW1 and AW2. Next, 20 µl elution buffer was applied and the column was spun for 1 min on a regular benchtop centrifuge at maximum speed. Recovery of the DNA sample in 20 µl of sterile water resulted in an average yield of ~3–5 ng/µl.

We used the SQK-RAD001 kit to prepare the DNA library. FRM (2.5 µl, ONT) was added to the DNA sample (20 µl) and incubated for 1 min at 30°C. Then, 1 µl RAD (ONT) plus 0.2 µl ligase was added and the mixture was incubated for 10 min.

The R9 flow cell was prepared by applying 500 µl priming mix (RBF 1x) twice. The library was then added to the flow cell without a purification step.

## Barcoding

The barcoding protocol was executed according to manufacturer's instructions for native barcoding kit I (EXP-NBD002, ONT) in conjunction with the Nanopore Sequencing kit (SQK-NSK007, ONT) with some modifications (*Supplementary file 1A*, exp. 4, exp. 5). In brief; 1.5 µg DNA was used as starting material for each sample and vigorously vortexed for 1 min. The DNA sample was end-repaired and dA-tailed using the NEBNext Ultra II End Repair/dA-Tailing Module (New England Biolabs [NEB] E7546S; 5 min 20°C, and 5 min 65°C). After an AMPure purification, the DNA fragments were subject to ligation using Blunt/TA Ligase Master Mix (NEB M0367S) for 5 min at 20°C and then 5 min at 65°C. The sample was then purified using AMPure magnetic beads and the DNA was eluted off the beads using 31 µl nuclease-free water. The NB01 and NB02 barcodes were ligated to the fragments of each sample with Blunt/TA Ligase mix (NEB) and incubated for 15 min. After an AMPure purification step, the two samples were pooled. Next, we ligated the adaptor (BAM) and hairpin (BHP) to the barcoded DNA fragments using NEB Quick Ligase (NEB) for 20 min at room temperature (22°C). The HTP (ONT) was added and incubated for another 10 min. The 50 µl MyOne C1 beads were prepared in the incubation step, which tethered the hairpin and ligated DNA fragments. The DNA library was eluted off the beads by ELB (ONT) at 37°C for 10 min and was applied to the flow cell.

## Reference databases

YE001, JP001 (https://dna.land/consent) and three HapMap samples (NA12890, NA12977, NA12879) are publicly available reference files. The 1,099 cancer cell line files were downloaded (GSE36139, CCLE), base-called using Birdseed and converted into 23andMe file format. The 31,000 DTC genomes were available from two sources: (i) 1,446 DTC genomes were downloaded from the public website OpenSNP.org and (ii) 29,554 genomes were collected using DNA.Land, an online website (https://dna.land). The website procedures were approved by our IRB. Based on current

**Table 1.** List of databases consulted and restrictions to access.

| Databases: | Restrictions to access | Dataset URL: |
|---|---|---|
| Opensnp.org | No | https://opensnp.org/ |
| HapMap[*] | No. The HapMap dataset has been discontinued (https://www.ncbi.nlm.nih.gov/variation/news/NCBI_retiring_HapMap/) and the archived HapMap data is available via FTP from ftp://ftp.ncbi.nlm.nih.gov/hapmap/. The relevant files used for this study have been downloaded from the latter in 2015. | http://www.completegenomics.com/documents/PublicGenomes.pdf and ftp://ftp.ncbi.nlm.nih.gov/hapmap/ |
| DNA.land | Yes. The 29,554 genomes provided by DNA.land are not available for distribution to ensure genomic privacy of the individuals who donated their genomes to DNA.land | https://dna.land/ |
| CCLE[†] | Yes. Public access is available by registration. The data made available on the Encyclopedia is for internal research purposes, as specified in CCLE Terms of Access (https://portals.broadinstitute.org/ccle/about). The SNP and Expression data from the Cancer Cell Line Encyclopedia (CCLE) is available on GEO under accession number GSE36139. | https://portals.broadinstitute.org/ccle/ and https://www.ncbi.nlm.nih.gov/geo/query/acc.cgi?acc=GSE36139 |

[*] *Drmanac et al., 2010*
† *Barretina et al., 2012*
DOI: https://doi.org/10.7554/eLife.27798.006

consent, this set of 29,554 genomes cannot be shared. All experiments with this collection were done using an automatic algorithm on a secure server without access to the explicit identifiers of the samples (e.g. names or contact information).

## MinION sketching

To start a MinION run, we primed the flow cell according to the manufacturer's protocol. We started MinKnow (protocol 'MAP_48 Hr_Sequencing_Run_SQK_MAP006' for R7 and 'NC_48 hr_Sequencing_Run_FLO-MIN104' for R9), uploaded the collected reads to Metrichor (a cloud-based program that base-called the reads), and stored them on our computer.

We used Poretools (*Loman and Quinlan, 2014*) to extract the FASTQ data and time stamps from the local files. Only reads with an average base quality greater than nine were used for the downstream analysis. Next, we aligned the files to hg19 using bwa-mem (v0.7.14) (*Li, 2013*) using the command 'bwa mem –V –x ont2d –t 4'. Reads with multiple alignments were not considered for further analysis.

To extract variants, we used a script to retain nucleotides from the MinION output that overlapped known positions of bi-allelic SNPs from dbSNP build-138 with an allele frequency between 1–99%. To minimize the effects of sequencing error, we considered only MinION read bases that matched the common SNP alleles in dbSNP. For example, if at position chr1:10,000 the MinION reported 'A' and dbSNP reported a variant 'C/G', then we treated this position as a sequencing error. The R7 chemistry run with NA12890 generated 4,920 variants after 1 hr of MinION sequencing, of which 7.7% were rejected after filtering for common SNPs. Intersecting these with the reference file and analyzing the true error from the matched SNPs resulted in 8.9% mismatches. This contrasted with the R9 chemistry, which only resulted in 2% true mismatches (*Supplementary file 1C-E*).

The Bayesian model was integrated in a Python script in order to match between the MinION data and each entry in the database. To accelerate the search, we implemented the following procedure: (i) if the posterior probability drops below $10^{-9}$, the script concludes that the database entry does not match and moves to the next entry, and (ii) the script uses only up to 1 hr of data to determine the posterior probability of a sample.

As a default setting, we used a prior probability of $10^{-5}$ for exact matching. The only exception was *Figure 2—figure supplement 1* (YE001), where we employed a range of prior probabilities. As a default setting, we used the computed error rate from each read as the ε in our Bayesian algorithm.

All code is publicly available on github at https://github.com/TeamErlich/personal-identification-pipeline (*Erlich, 2017*). A copy is archived at https://github.com/elifesciences-publications/personal-identification-pipeline.

## Simulations

For the simulations, we took reads from exp. 4 and 5 (*Supplementary file 1A*). The total number of reads was set to 3,000 and a random number of reads that represented the required proportion were selected. For example, for 50% contamination, we took 1,500 random reads from exp. 4 and 1,500 random reads from exp. 5. These were pooled together and again shuffled to simulate a mixture. This process was repeated five times for each contamination fraction. The resulting pooled file was processed using our pipeline and matched to the reference file of the corresponding MinION sketch (either THP1, or JP001).

## Declarations

Availability of the data: The code for our method is available on https://github.com/TeamErlich/personal-identification-pipeline (*Erlich, 2017*). A copy is archived at https://github.com/elifesciences-publications/personal-identification-pipeline. Replicating the experiments can be done using the THP1 test example (all data available on Github). The MinION reads for THP1 are also available in Zenodo at https://zenodo.org/record/1035914; 10.5281/zenodo.1035913). Building databases for your own MinION queries can be done by using opensnp.org, for cancer cell lines by downloading the CCLE reference files or using your own private SNP array files relevant to your query. The 29,554 genomes provided by DNA.land are not available for distribution to ensure genomic privacy of the individuals who donated their genomes to DNA.land (see Materials and methods section: Reference databases and *Table 1*).

## Acknowledgements

We thank Eleazar Eskin for useful comments on the model. We thank Neville Sanjana for providing cell lines and for discussions. We thank Jae Young Choi, Thomas Willems and Dina Zielinski for useful comments, William Stephenson and Kunal Pandit from New York Genome Center's Innovation lab for technical assistance and Aaron Wolman for providing the THP1 cell line. We thank Michael Micorescu from Oxford Nanopore Technologies for useful discussions, and the Columbia Ubiquitous Genomics class 2015 for data generation.

## Additional information

### Competing interests

Yaniv Erlich: YE is a consultant for DNA forensics company ArcBIO and co-founder of DNA.land. The other authors declare that no competing interests exist.

### Funding

| Funder | Grant reference number | Author |
| --- | --- | --- |
| Burroughs Wellcome Fund | | Yaniv Erlich |
| National Institute of Justice | 2014-DN-BX-K089 | Yaniv Erlich |
| Andria and Paul Heafy | | Yaniv Erlich |

The funders had no role in study design, data collection and interpretation, or the decision to submit the work for publication.

### Author contributions

Sophie Zaaijer, Conceptualization, Formal analysis, Supervision, Validation, Investigation, Visualization, Methodology, Writing—original draft, Writing—review and editing; Assaf Gordon, Resources, Software, Supervision, Visualization; Daniel Speyer, Methodology; Robert Piccone, Software; Simon Cornelis Groen, Formal analysis, Writing—review and editing; Yaniv Erlich, Conceptualization, Software, Formal analysis, Supervision, Funding acquisition, Validation, Investigation, Visualization, Methodology, Writing—original draft, Writing—review and editing

## Author ORCIDs
Sophie Zaaijer (iD) http://orcid.org/0000-0002-0437-8620
Simon Cornelis Groen (iD) http://orcid.org/0000-0003-4538-8865
Yaniv Erlich (iD) http://orcid.org/0000-0003-3257-8387

## Ethics

Human subjects: All individuals (YE001, JP001, SZ001) declare they fully consented to participate in the study and to the publication of the MinION data. Further information can be found in the Materials and methods section subheading: "Reference databases'.

## Decision letter and Author response

Decision letter https://doi.org/10.7554/eLife.27798.018
Author response https://doi.org/10.7554/eLife.27798.019

# Additional files

## Supplementary files

• Supplementary file 1. Supplementary Tables. Run statistics for the MinION sketch experiments.
DOI: https://doi.org/10.7554/eLife.27798.014

• Transparent reporting form
DOI: https://doi.org/10.7554/eLife.27798.015

## Major datasets

The following dataset was generated:

| Author(s) | Year | Dataset title | Dataset URL | Database, license, and accessibility information |
|---|---|---|---|---|
| Sophie Zaaijer, Assaf Gordon, Daniel Speyer, Robert Piccone, Simon Cornelis Groen, Yaniv Erlich | 2017 | Personal Identification Pipeline - THP1 Sample | https://dx.doi.org/10.5281/zenodo.1035914 | Publicly available at Zenodo (https://zenodo.org/). |

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
