## [Decision Letter]

[Editors’ note: this article was originally rejected after discussions between the reviewers, but the authors were invited to resubmit after an appeal against the decision.]

Thank you for submitting your work entitled "Rapid DNA Re-Identification for Cell Line Authentication and Forensics" for consideration by *eLife*. Your article has been reviewed by three peer reviewers, one of whom is a member of our Board of Reviewing Editors, and the evaluation has been overseen by a Senior Editor. The following individuals involved in review of your submission have agreed to reveal their identity: Nicholas Loman (Reviewer #3).

Our decision has been reached after consultation between the reviewers. Based on these discussions and the individual reviews below, we regret to inform you that your work will not be considered further for publication in *eLife*.

You will see from the reviewers comments below that they agreed that the methodology to identify individuals or cell lines using very low coverage MinION data, assuming the availability of an existing reference database, is sound and that the results of the application of the approach to individual and cell line identification provided a convincing theoretical demonstration of the technique. However, the reviewers raised substantial concerns about the practical utility of the approach in forensics and cell line identification because, for example, of the lack of availability of a reference and the impact of contamination. Given that the emphasised applications are a central element the paper, the reviewers recommended considering other applications, potentially outside the field of human genetics, that would more broadly increase the impact of the technique.

Reviewer #1:

The authors present a novel approach, "MinION sketching" to enable rapid, inexpensive and portable human DNA re-identification. The authors suggest that their approach requires 3 minutes and "91 random SNPs" to identify a sample, and demonstrate applications to cell line authentication or forensics.

Major comments:

I have no concerns about the statistical approach, but it wasn't clear to me how the performance of the algorithm depended on the properties of the SNPs used. With common SNPs, there will be greater probability of a genotype match with a random DNA sample? But presumably common SNPs are more likely to be reported in relevant databases? Is the database of 31,000 individuals likely to be representative of the DNA databases used in forensics (which I still thought were predominantly based on STRs rather than SNPs)? How much does the approach depend on the number of individuals and numbers of SNPs in the reference? Similarly, with the cell line authentication, what impact does the use of a different technology in the reference make on utility and the interpretation of findings? I also didn't see any discussion of "stopping rules" – how does the method determine there is sufficient evidence that there is no match (since the prior probability of a match will be small)?

Whilst the algorithm is currently defined as providing probabilities of a match, can the approach be extended to provide probabilities that the samples are from close relatives?

The novel approach is claimed to be rapid and inexpensive, but is the need for "rapid" response so important in the applications of forensics or cell line authentication? How does the novel approach compare in terms of the amount of DNA required compared with currently used techniques, and does this reduce utility?

Reviewer #2:

This manuscript described the use of the commercially available MinION system based on sequencing via nanopore technology to analyse a limited number of 4 (in words: four) DNA samples in various technical scenarios. The authors then compared the MinION outcome with a SNP microarray database they collected from Direct-to Consumer companies and other sources, to establish sample matches using a Bayesian algorithm they developed. They claim that besides the relatively high error rate of MinION, their algorithm provides evidence for sample matching, likely because of the large number of SNPs generated by both approaches. Based on their results, the authors suggest the future practical use of this approach for re-identification purposes in forensics and elsewhere, and advocate its advantage of a cheap, quick and PCR-free approach, which contrasts to the more expensive, slower PCR approach currently used such as in forensics. Besides it being unclear what the advantage is to speed-up the re-identification for the price of using an error-prone sequencing device (which in turn needs a statistical approach to compensate for the errors), it also is unclear why a PCR-free approach shall provide an advantage for these type of data. Clearly, STR typing benefits from avoiding PCR as this eliminates disturbing slippage artifacts, which can trouble re-identification; however, SNPs do not generate slippage artifacts. Obviously, general advantage of using PCR for identification purposes is that minute amounts of DNA can be successfully used, which is especially suitable to forensic identification. However, I cannot find in the manuscript the sensitivity limits of this approach, which I expect to be higher (i.e. more DNA needed) than possible when using PCR, which would limit the application. Moreover, when proposing the use of their approach in forensics, the authors seem to ignore that SNPs are not used in routine forensic DNA analysis despite their technical advantages of avoiding slippage artifacts etc. simply because forensic reference databases consist of STRs. Hence, even by ignoring the various caveats of this approach when it comes to the robustness and reliability of a forensic DNA test, which I cannot see validated in this study, the usefulness of their approach for identification purposes in forensics as suggested by the authors does not exist in practice. And even if forensic DNA databases would ever move to SNPs, which has been discussed for many years but not a single country has adapted this, the proposed approach would be not suitable, because it would require that reference samples and trace samples would be analysed with different technologies, which causes additional complication that are avoided by the use of the same markers and technology for both type of forensic samples, as is currently the case. Whether their developed Bayesian approach that allows matching error-prone MinION data with error-poor SNP microarray data, which to me is the heart of this manuscript but not its application, is technically sound and novel enough to justify publication in a high-profile life science journal, escapes this reviewer's technical background knowledge and shall be evaluated by a statistical genetics expert instead; the proposed application of their approach for re-identification purposes at least in the field of forensics does not. Another statistical genetics issue that shall be evaluated by a respective expert is if the number of SNPs matched between MinION and reference array dataset, as achieved with this approach, is truly enough for statistically sound individual identification, which not only depends on SNP numbers but also on degree of variation. However, from what I can see I expect that the SNPs matching between both datasets appear to be different between individuals, this issue would require a careful evaluation using much more data then presented here.

Reviewer #3:

This manuscript details a computational framework and an experimental design for the rapid identification of human subjects employing low coverage, noisy nanopore reads. It builds on existing work previously published in *eLife* relating to using nanopore sequencing in the classroom (https://elifesciences.org/articles/14258).

The method and results are solid and convincing as a method to identify individuals or cell lines using very low coverage MinION data, assuming the availability of an existing reference database.

The Bayesian method used is elegant and seems to give good results, demonstrating the power of combinations of unlinked SNPs in generating a unique signature of humans. We previously demonstrated a similar real-time streaming identification approach for pathogens using phylogenetic placement, a slightly different but related approach (https://genomebiology.biomedcentral.com/articles/10.1186/s13059-015-0677-2)

The ability to rapidly identify individuals outside of a traditional laboratory environment using forensic DNA sequencing will be interesting to a broad audience including biologists, forensic scientists and the public. The authors previously termed this 'CSI' sequencing and I think that is the source of the appeal of this idea and manuscript.

My main concerns about this work relate mainly to whether it has genuine practical uses.

The first example – of human identification seems limited in utility for the following reasons:

1) A reference database, generated from whole-genome sequences or genotyping panels is required for this to work. (And the data generated by this technique cannot be used to populate such a database, meaning a parallel reference database building effort must be employed).

2) The database used for this work from the DNA.land website is not publicly available (nor can it be for privacy reasons), to permit others to reproduce this work.

3) Even if it was – should work like this be encouraged? I find it a little hard to think of outside of forensic investigations where the ability to identify people from their DNA would be something routinely practiced.

In the absence of large scale genetic databases it is hard to see how this could be useful. Would the authors advocate the collection of identifiable large scale genetic databases by authorities? How would this work in practice? If not, is this purely a theoretical demonstration?

The second example given – that of cell line identification – does seem like a potential practical use, although at present the cost of such analysis would likely hinder its adoption compared with a simpler STR panel based approach for identification. Although the authors state that this is done by sending samples off and at great cost, this is also available to researchers to run in their own labs at not great cost (GenePrint system from ProMega at <$10/rxn). The authors may want to discuss how the cost could be brought down to similar levels.

The method as shown is also not likely to work well with lower levels of contamination and/or from contamination from multiple cell lines and does not seem to provide identification in such mixtures.

In summary, I think to improve this article the authors should really spend time outlining the potential practical uses of this technique (including outside of human genetics) and then discuss in more detail the ethical concerns associated with such uses.

[Editors’ note: what now follows is the decision letter after the authors submitted their appeal.]Thank you for choosing to send your work entitled "Rapid DNA Re-Identification for Cell Line Authentication and Forensics" for consideration at eLife. Your letter of appeal has been considered by a Senior Editor and a Reviewing editor, and we are prepared to consider a revised submission with no guarantees of acceptance. We encourage such dialogue post decisions, and welcome the chance to review those decisions. The reviewers were more convinced by the potential application to issues of cell line contamination after reading your letter of appeal, and would recommend that you include all the points made in your rebuttal in the revised version of the manuscript. The reviewers remain less convinced about the application for forensic human identification, and suggest including much more detail in the discussion of the limitations of the approach in this context and the barriers for putting it into practice, whilst speculating that forensics might move to using SNPs in the future.

[Editors’ note: what now follows is the decision letter after the authors submitted for further consideration.]

Thank you for resubmitting your work entitled "Rapid DNA Re-Identification for Cell Line Authentication and Forensics" for further consideration at *eLife*. Your revised article has been favorably evaluated by Mark McCarthy (Senior editor), a Reviewing editor, and two reviewers.

The manuscript has been improved but there are some remaining issues that need to be addressed before acceptance.

The reviewers remain unconvinced that the proposed method is suitable for the needs and requirements of forensic DNA analysis, as outlined in our original rejection letter. The reviewers appreciated the increased emphasis on cell line authentication in the revised version of the manuscript, for which the method is more appropriate, but are not satisfied that forensic applications have been sufficiently de-emphasised, as requested in our response to the authors' letter of appeal.

The reviewers particularly commented that rapid and mobile DNA analysis is not applied for standard DNA profiling for many reasons, including sensitivity, genotype quality, and non-controllable environment increasing risks of contamination, which would also apply to the proposed method. The proposed method is not applicable to STRs, on which almost all forensic DNA analyses are based (including all existing forensic DNA databases), and is not suitable for the analysis of low quantity DNA typically available from crime scene stains.

To be acceptable for publication, the reviewers have requested the following changes be made:

1) Remove "and forensics" from the title.

2) Remove "or in some forensic applications" from the title.

3) Remove from the Introduction the fourth paragraph describing forensics.

4) Remove the paragraphs from the Discussion that discuss the utility of the approach for forensics, i.e. the first two paragraphs of the subsection “Forensics”.

5) Remove "and to provide an alternative method for DNA-based forensics" from the concluding remarks.

In addition, please address the following issues:

1) The reference database (http://files.teamerlich.org/pidp/CCLE_genotypes.tar.gz) should be deposited in a public repository (not the lab website).

2) Concern has been raised over the availability of the code, which should be made publicly available by releasing it with some kind of Creative Commons license, for example.

3) The competing interest statement (Y.E. is a consultant for a DNA forensic company) should be more explicit.

---

## [Author Response]

[Editors’ note: the authors’ letter of appeal in response to the first round of peer review follows.]

You will see from the reviewers comments below that they agreed that the methodology to identify individuals or cell lines using very low coverage MinION data, assuming the availability of an existing reference database, is sound and that the results of the application of the approach to individual and cell line identification provided a convincing theoretical demonstration of the technique. However, the reviewers raised substantial concerns about the practical utility of the approach in forensics and cell line identification because, for example, of the lack of availability of a reference and the impact of contamination. Given that the emphasised applications are a central element the paper, the reviewers recommended considering other applications, potentially outside the field of human genetics, that would more broadly increase the impact of the technique.

Many thanks for your thorough reading of our paper and for the reviewers’ comments. We are very happy that all reviewers unanimously agreed our method is sound and convincing. However, we were surprised to find that it was also concluded that there is a lack of (potential) applications. We feel strongly that there is a suite of highly impactful applications of our method, which could only expand in the future. We fear that we have not been able to convey this sufficiently in our manuscript, and we hope that you would reconsider your decision based on the arguments that we outline below:

Cell line contamination is a long-standing and persistent problem in academic, clinical and commercial research. The available solutions for cell line authentication are clearly not sufficient, since contamination is still a major economic burden on society despite having been recognized nearly 50 years ago. Issues with cell line contamination and authentication are very well described by Almeida et al 2016 PLOS Biology. We would like to explain a couple of points raised:

1. Contamination and rapid identification: cell line authentication benefits from being able to identify sample contamination in the earliest stage possible. The current US national guidelines by the ASN0002 state that a cell line is pure if the STRs have a match of 80% to one of the entries in the database. Only when the STR match < 56% the cell line is considered contaminated. Our method is better and more applicable than existing methods in two crucial aspects:

1. Cell line authentication is not a static one-off test: Transfer of a couple of cells from one culture to the other by human error happens. Only the cells that rapidly proliferate compared to the original cell line will jeopardize the research done over time. Therefore the *periodic testing* of all cell lines in the lab is crucial, since only periodic testing will reveal that a cell line is contaminated. The *ATCC takes two weeks* to return test results, which means that by the time a researcher gets his/her results back, the contaminant might have overgrown the culture. Our MinION sketching method provides a rapid, and local solution for periodic cell line testing to allow researchers to be fully aware of the state of their cell line(s) at the time of an experiment. Even with a contamination of 10% our method rarely provides a robust match, indicative of contaminants in the cell line. Plus, we can robustly pick up a contamination of 20%, which is an important two-fold improvement over the ASN0002 standard that only rejects a pure cell line when < 56% matches.

2. Sample mix-up: swapping cell lines does happen in laboratories from time to time and this would be a 100% contamination event. The MinION sketch would provide a perfect, instant sanity check prior to starting an experiment for any researcher.

2. Contamination analysis using SNPs versus STRs: cancer cell lines are typically very unstable and proliferate rapidly. STRs are more likely to incur mutations than SNPs and therefore, even a pure cancer cell line typically will result in a non-homogenous mix of repeat lengths. Hence, the ASN0002 guidelines place a cut off at an 80% match for pure cancer cell lines. Our SNP-based MinION sketch therefore enables a higher accuracy for the detection of contaminations in cancer cell lines.

3. The reference database availability: the cancer cell line encyclopedia generated by the Broad Institute provides around a ~1,099 reference files for the general public to use. This is only the start, and the database will be updated as the number of cell lines available will increase. When our method is used for forensic purposes ethical issues need to be considered – however, the appropriate (government) bodies that regulate forensic research and investigations should be able to formulate regulations that allow for the use of our method alongside relevant reference databases (see below) in forensics.

4. Cost: The costs of sequencing are continuing to plummet and this trend will most probably extend into the near future. When samples are multiplexed using 12 barcodes the costs of our MinION sketching method is currently already comparable to, and even lower than, the available cell line authentication method via the ATCC. The difference between our MinION sketching method versus the ATCC method is that researchers do not have to wait for two weeks to obtain the test results.

5. DNA concentration: for cell line authentication we do not have to worry as much about the availability of the input DNA as the forensic field has to. Even without pushing the boundaries of the possible, we were able to identify an individual using 50ng DNA. Our method can be used in addition with a PCR amplification step of informative SNPs, which could be an expansion of the method in the situation of an extremely low input DNA sample. We do concede that this would add a time consuming step in the protocol, however.

6. Expansion to other fields: Our method might very well be suitable for other fields, such as mice cell line authentication, and for testing the purity of *C. elegans*, Drosophila, plant and even yeast stocks. This paper can be the first step to make purity testing more convenient for researchers from disciplines across the life sciences – this is the reason why we think this paper would be perfect for eLife with its broad readership.

Forensic applications: We know the field of STR research very well and the Erlich lab has published several key papers in this area (e.g. Gymrek et al., Nature Genetics, 2016). We fully agree with reviewer 2 that the forensic community will not replace their STR CODIS system just because of our manuscript.

Having said that, there does seem to be quite a bit of excitement about our MinION

sketching method in the forensic community. The National Institute of Justice (NIJ) funded our work, and the American Academy of Forensic Sciences (AAFS) invited us to present this work in the session on DNA forensics in their 2016 annual meeting in Las Vegas. Following our presentation at the AAFS meeting, we were contacted by a group of researchers from the DNA forensic team of Battelle, one of the largest government contractors in this area. They were interested in testing our method with their samples for a special report to the Department of Defense (DoD). We were also invited to present this work in the Oxford Nanopore Technologies group meeting and received requests from NYU and German collaborators to use our method for portable field re-identification of crop and animal stocks.

We do not envision that our method will compete with the STR method for regular forensic case work in the near future. However, we do see situations such as mass disasters at the scale of 9/11, the 2004 tsunami, or the Tōhoku earthquake that will require identification of a large number of casualties in circumstances where moving tissues to labs is logistically complicated. MinION sketching would allow for the rapid, on-site generation of DNA signatures using an inexpensive, off-the-shelf portable device. There are also other forensic tasks, especially at the DoD, that will benefit highly from MinION sketching.

We apologize that the text of our manuscript was not clear enough to describe the benefits of our method and our motivations behind its development. In the light of the information above, would you be able to consider giving us a chance to enhance the clarity of our text, address all reviewers comments and re-submit?

[Editors’ note: the authors’ point-by-point response to the first round of peer review now follows.]

Reviewer #1:I have no concerns about the statistical approach, but it wasn't clear to me how the performance of the algorithm depended on the properties of the SNPs used. With common SNPs, there will be greater probability of a genotype match with a random DNA sample? But presumably common SNPs are more likely to be reported in relevant databases? Is the database of 31,000 individuals likely to be representative of the DNA databases used in forensics (which I still thought were predominantly based on STRs rather than SNPs)? How much does the approach depend on the number of individuals and numbers of SNPs in the reference? Similarly, with the cell line authentication, what impact does the use of a different technology in the reference make on utility and the interpretation of findings? I also didn't see any discussion of "stopping rules" – how does the method determine there is sufficient evidence that there is no match (since the prior probability of a match will be small)?

Many thanks for these great questions. We will go through them one by one, and in the paper we also clarified the points raised.

SNPs:

In matching two DNA profiles (MinION sketch and SNP array data) our algorithm considers the likelihood of a match and no match (i.e. the probability of a random match). For example, for the likelihood of a “match”: if at position chr1:10,000 the MinION reported “C” and dbSNP reported a variant “C/G”, then the match likelihood gets the value: 1-(error rate).

For the likelihood of “no match” the allele frequency is considered. Let’s say the sequenced C at that position is seen very frequently in the population (e.g. 90% of individuals in the population have a C at chr1:10,000). In that case we consider that likelihood of the “no-match” (how likely would we see this allele in the population if we would randomly sequence an individual?). On the other hand, if this variant had been a G, we would have gained a lot of information, as one would not be likely to encounter that sequenced variant randomly. You can find this reflected in the denominator of our algorithm in our Materials and methods section. In the Materials and methods section we describe all details of the model in more depth. Please note, we only use common SNPs with allele frequencies between 1-99%.

Database:

Most databases for forensics and also many cell line databases are based on STRs. Forensic databases have millions of DNA profiles (https://www.fbi.gov/services/laboratory/biometricanalysis/codis/ndis-statistics). Indeed, those databases are not compatible with our MinION sketch method. However, for research and clinical purposes cell lines and tissue samples are commonly genotyped using SNP arrays or WGS, and these data reference files *are* compatible with MinION sketching (an example: the CCLE reference files we provide in this manuscript). In forensics, people are still hesitant to use SNP-based databases, yet, specific applications like identifying victims from mass disasters might benefit from SNP array reference files that have been generated either by law enforcement bodies or DTCs. A growing number of individuals are already releasing their reference files online via openSNP, and this publically available database is growing. The DNA.land database is not publically available, but served as a proof of principle database to benchmark our method.

Database size: our method matches a MinION sketch to a single reference file in the database at a time and computes the posterior probability that there is a match. These calculations are independent of the size of the database used. However, the larger the database the larger the chance that the reference file that should match to a MinION sketch is present in the database. We do have a prior that reflects the search space – in Figure 2—figure supplement 1 present in the dat we explore the effect of changing the prior, and find that our method is robust even if we assume the whole world population (7.5*10^[7]^).

Number of SNPs in a reference file: As an example; running the MinION for one hour (for SZ001) resulted in the isolation of 13,453 common variants. Intersecting those with her reference file (560K SNPs) only 457 variants intersected. Of course, each MinION run results in different yields. So the more reads that pass through the nanopores per unit of time, the higher the number of variants that are sketched and intersected with the reference files, and the faster sample reidentification can be achieved. And indeed, a higher number of SNPs in the reference files results in an increased chance to intersect SNPs from a MinION sketch.

Thanks very much for bringing up the “stopping rules”. We added an analysis to find the minimum number of SNPs to infer the match. (Figure 4). Intersecting 300 SNPs was sufficient to find the correct matching reference file for 99.6% of the MinION sketches we simulated. Going past that number of SNPs without observing a posterior probability that does not reach and stay at 99.9% means the correct matching reference file is most likely not in the database.

Whilst the algorithm is currently defined as providing probabilities of a match, can the approach be extended to provide probabilities that the samples are from close relatives?

This is an excellent point. In this manuscript we wanted to establish a robust method for inferring an exact match. Even a first-degree relative is not identified as a candidate match using our method, which was an important test to validate our method. We completely agree that familial matching would be a valuable enrichment of the search space, and we are therefore currently working on this expansion of the method. However, making our algorithm robust and suitable for inferring familial matches requires major additions, and we feel that this would merit its own manuscript. On top of this, while familial matching would potentially open up a suite of additional applications in forensics, it would not add to the applicability of our method in the authentication of cell lines, and tissues in biobanks and the clinic. Adding this in would dilute our ability to highlight major issues with authentication that cause long-standing and ongoing problems in the reproducibility of research and for which our method can be an important part of the solution.

The novel approach is claimed to be rapid and inexpensive, but is the need for "rapid" response so important in the applications of forensics or cell line authentication? How does the novel approach compare in terms of the amount of DNA required compared with currently used techniques, and does this reduce utility?

We thank the reviewer for bringing this up, so we get the opportunity to clarify our reasoning.

Why do we need rapid cell line authentication?

1) Cell lines and their potential contaminants are living and dynamic entities. Two weeks in culture can change the composition of a cell culture dramatically. We added a simulation analysis in the manuscript that shows this effect over time (Figure 5—figure supplement 1). It is crucial to be able to monitor the genomic make-up of a population of cells in culture periodically, so that researchers can be confident that all experiments are done with clean and verified cell lines. We added sections in the Discussion that elaborate more on this point.

2) Since periodic testing is a necessity whenever one is working with biological materials and cell lines in particular, the re-identification method needs to be easy and rapid to execute for researchers or clinicians. The main reason for the long-standing problems with cell line authentication is human behavior. Nobody has time. Therefore providing a method that can be done in an hour will potentially go a long way in solving the issue of reluctance to test cell lines.

Why do we need rapid re-identification in forensics?

Currently, DNA profiling for forensic purposes suffers from the latency it takes to transport and test a sample, in addition to the availability of appropriate equipment. A huge backlog of sample processing is known to exist for profiling DNA from, for example, rape kits, which is, among other things, due to a lack of available manpower and equipment

(http://www.endthebacklog.org/backlog/what-rape-kit-backlog). The delay in testing means that in the meantime the perpetrator is free to commit further crimes.

For crime scenes, MinION sketching could be a first check on-site to be able to act rapidly and prevent further crime.

Cost is mentioned as an issue in forensics; yet, a low-cost device like the MinION could potentially be distributed among a larger proportion of the police corps and be operated by lightly trained police officers. This could help to increase the efficiency of police work. Are we fully there already? Not yet, we need devices like the "Zumbador" for DNA extraction and sample preparation (in development at ONT) to be able to do on-site re-identification of DNA samples with minimal effort. Further studies will also enable identification of mixed samples. Large-scale adoption would require a drop in cost per flow cell. Even so, our method could contribute to yet another major step towards a new system where no DNA sample in forensics is left untested.

DNA concentration:

For cell line authentication DNA concentration is not an issue, in most situations enough DNA material is present.

For forensics DNA concentration can be an important limiting issue indeed. A whole genome amplification step for the sparse genomic regions found in the sample can be added to the library preparation protocol for MinION sketching. Particularly in these cases we think that not having to rely on specific sites, but being able to randomly sample SNPs from across the genome improves the chance on finding a match if the correct matching file is present in the reference database used.

Reviewer #2:This manuscript described the use of the commercially available MinION system based on sequencing via nanopore technology to analyse a limited number of 4 (in words: four) DNA samples in various technical scenarios. The authors then compared the MinION outcome with a SNP microarray database they collected from Direct-to Consumer companies and other sources, to establish sample matches using a Bayesian algorithm they developed. They claim that besides the relatively high error rate of MinION, their algorithm provides evidence for sample matching, likely because of the large number of SNPs generated by both approaches. Based on their results, the authors suggest the future practical use of this approach for re-identification purposes in forensics and elsewhere, and advocate its advantage of a cheap, quick and PCR-free approach, which contrasts to the more expensive, slower PCR approach currently used such as in forensics. Besides it being unclear what the advantage is to speed-up the re-identification for the price of using an error-prone sequencing device (which in turn needs a statistical approach to compensate for the errors), it also is unclear why a PCR-free approach shall provide an advantage for these type of data.

We thank the reviewer for bringing this up, so we get the opportunity to clarify our reasoning.

Why do we need rapid cell line authentication?

1) Cell lines and their potential contaminants are living and dynamic entities. Two weeks in culture can change the composition of a cell culture dramatically. We added a simulation analysis in the manuscript that shows this effect over time (Figure 5—figure supplement 1). It is crucial to be able to monitor the genomic make-up of a population of cells in culture periodically, so that researchers can be confident that all experiments are done with clean and verified cell lines. We added sections in the Discussion that elaborate more on this point.

2) Since periodic testing is a necessity whenever one is working with biological materials and cell lines in particular, the re-identification method needs to be easy and rapid to execute for researchers or clinicians. The main reason for the long-standing problems with cell line authentication is human behavior. Nobody has time. Therefore providing a method that can be done in an hour will potentially go a long way in solving the issue of reluctance to test cell lines.

Why do we need rapid re-identification in forensics?

Currently, DNA profiling for forensic purposes suffers from the latency it takes to transport and test a sample, in addition to the availability of appropriate equipment. A huge backlog of sample processing is known for profiling DNA from, for example, rape kits, which is, among other things, due to a lack of available manpower and equipment

(http://www.endthebacklog.org/backlog/what-rape-kit-backlog). The delay in testing means that in the meantime the perpetrator is free to commit further crimes.

For crime scenes, MinION sketching could be a first check on-site to be able to act rapidly and prevent further crime.

Cost is mentioned as an issue in forensics; yet, a low-cost device like the MinION could potentially be distributed among a larger proportion of the police corps and be operated by lightly trained police officers. This could help to increase the efficiency of police work. Are we fully there already? Not yet, we need devices like the "Zumbador" for DNA extraction and sample preparation (in development at ONT) to be able to do on-site re-identification of DNA samples with minimal effort. Further studies will also enable identification of mixed samples. Large-scale adoption would require a drop in cost per flow cell. Even so, our method could contribute to yet another major step towards a new system where no DNA sample in forensics is left untested.

Why is a PCR-free approach an advantage? Our reasoning is the following:

First, having to perform PCR adds yet another step in the protocol that takes time.

Second, the current protocols mostly use human-specific PCR primers and therefore contaminants from other species cannot be detected [Alston-Roberts, Nature reviews 2010].

Third, cancer cell lines are genetically unstable, and loss of heterozygosity or microsatellite instability result in a reduction in matching precision. Whether one chooses to amplify the DNA with a whole genome amplification approach or to use no PCR at all, either approach can be part of a robust re-identification method since there is no need to rely on a limited number of specific bi-allelic sites. The reliance on a small set of markers makes a reidentification method sensitive to allelic dropouts.

Clearly, STR typing benefits from avoiding PCR as this eliminates disturbing slippage artifacts, which can trouble re-identification; however, SNPs do not generate slippage artifacts. Obviously, general advantage of using PCR for identification purposes is that minute amounts of DNA can be successfully used, which is especially suitable to forensic identification. However, I cannot find in the manuscript the sensitivity limits of this approach, which I expect to be higher (i.e. more DNA needed) than possible when using PCR, which would limit the application.

DNA concentration:

For cell line authentication DNA concentration is not an issue, in most situations enough DNA material is present.

For forensics DNA concentration can be an important limiting factor. A whole genome amplification step for the sparse genomic regions found in the sample can be added to the library preparation protocol for MinION sketching. Particularly in these cases we think that not having to rely on specific sites, but being able to randomly sample SNPs from across the genome improves the chance on finding a match if the correct matching file is present in the reference database used.

Moreover, when proposing the use of their approach in forensics, the authors seem to ignore that SNPs are not used in routine forensic DNA analysis despite their technical advantages of avoiding slippage artifacts etc. simply because forensic reference databases consist of STRs. Hence, even by ignoring the various caveats of this approach when it comes to the robustness and reliability of a forensic DNA test, which I cannot see validated in this study, the usefulness of their approach for identification purposes in forensics as suggested by the authors does not exist in practice.

We apologize for not stating this clearly in the text. We added to the discussion hopefully clearer pointers to highlight the importance of suitable reference databases and the fact that current databases are STR based.

The error-rate of the MinION is too high to sequence STRs at the moment, for example please see: https://nanoporetech.com/publications/2015/08/04/sequencing-analysing-and-counting-shorttandem-repeats-on-the-minion-2. We did not feel like an additional experiment benefits the manuscript to showcase this further.

A direct comparison of the statistical power using STR versus SNP markers for re-identification is done by (among others) Sanchez et al., 2010 and Yu et al., 2015, in the discussion we explain this further.

And even if forensic DNA databases would ever move to SNPs, which has been discussed for many years but not a single country has adapted this, the proposed approach would be not suitable, because it would require that reference samples and trace samples would be analysed with different technologies, which causes additional complication that are avoided by the use of the same markers and technology for both type of forensic samples, as is currently the case.

In forensics one would ideally use the same device for the re-identification and for generating the profile to make the database building efficient logistically. Yet, what has the highest priority; the fact that you could potentially re-identify DNA rapidly and on-site, or the efficiency of building a database system? This question would need to be considered for each application opportunity (perhaps not for regular crime scene testing right now, but after mass disasters new reference databases might have to be created which benefit from being SNP based, as people can contribute their DTC generated files for example – the purpose here is to find one's loved one, and people will be willing to help). Our method might open up new opportunities.

For the application of cell line authentication, we show in this manuscript that we can robustly use SNP arrays in conjunction with the MinION sketch to re-identify a sample. This has the particular application to test the cell line after arrival in labs and for the crucial sanity check, or after extensive passaging. For this application there does not seem to be a clear advantage of building the database and re-identification by the same device.

Whether their developed Bayesian approach that allows matching error-prone MinION data with error-poor SNP microarray data, which to me is the heart of this manuscript but not its application, is technically sound and novel enough to justify publication in a high-profile life science journal, escapes this reviewer's technical background knowledge and shall be evaluated by a statistical genetics expert instead; the proposed application of their approach for re-identification purposes at least in the field of forensics does not. Another statistical genetics issue that shall be evaluated by a respective expert is if the number of SNPs matched between MinION and reference array dataset, as achieved with this approach, is truly enough for statistically sound individual identification, which not only depends on SNP numbers but also on degree of variation. However, from what I can see I expect that the SNPs matching between both datasets appear to be different between individuals, this issue would require a careful evaluation using much more data then presented here.

We thank the reviewer for elaborating on these points. We modified the manuscript and hope it conveys our message more clearly and convincingly.

First, we redirected the focus of our manuscript to an immediately implementable application of our method, namely cell line authentication. Our method can be done very rapidly, on-site and can be adopted by labs right away. This will hopefully contribute to a major stride forward in reducing the use of unknown cell lines.

Second, for the forensics field:

Multiple studies have shown that the statistical power of 48-52 SNPs is comparable to the current 10-15 STRs used (for example: Sanchez et al., 2010 and Yu et al., 2015. Moreover, Lin et al., 2004 showed in a theoretical framework that ~80 random SNPs are sufficient to discriminate individuals from each other. Our study shows that we indeed re-identify DNA using only 60-300 SNP markers. The reason for our range is A) we randomly sample from the genome and encounter variants that are more informative than others due to their allele frequency. This is an advantage, as we don’t rely on specific sites in the genome, and we therefore do not suffer from allelic dropouts. B) We have to work with a relatively high error-rate. Even so, we show in our manuscript that we robustly identify individuals without exceptions.

We acknowledge some major hurdles still need to be taken before this method can be implemented in the field. However, doesn’t the reviewer agree there is space to improve our forensics methodologies in terms of speed and efficiency? This manuscript suggests a solid way to do so, using cutting edge technology, and proposes yet another step forward to make this happen.

Reviewer #3:[...]My main concerns about this work relate mainly to whether it has genuine practical uses. The first example – of human identification seems limited in utility for the following reasons:1) A reference database, generated from whole-genome sequences or genotyping panels is required for this to work. (And the data generated by this technique cannot be used to populate such a database, meaning a parallel reference database building effort must be employed).2) The database used for this work from the DNA.land website is not publicly available (nor can it be for privacy reasons), to permit others to reproduce this work.3) Even if it was – should work like this be encouraged? I find it a little hard to think of outside of forensic investigations where the ability to identify people from their DNA would be something routinely practiced.In the absence of large scale genetic databases it is hard to see how this could be useful. Would the authors advocate the collection of identifiable large scale genetic databases by authorities? How would this work in practice? If not, is this purely a theoretical demonstration?

Many thanks for the thoughtful introduction and the positive feedback.

DNA.land files are not published for reasons of protecting the privacy of the users – however please note that this database includes 1,446 reference files available on OpenSNP.org (described in the Materials and methods section). These files can be downloaded and used as a database to reproduce our results. Each forensic body will have to further build their own (privacy protected) database, similar to the CODIS system currently employed. Since the use of SNP arrays and WGS in genotyping patients is becoming standard practice in biobanks and hospitals as part of the ‘personalized medicine’ efforts, each institution will essentially (have) create(d) its own reference database that can be used for sample authentication in house, when necessary.

Furthermore, our work can certainly be reproduced:

The genomes of YE001, JP001 and the HapMap samples (NA12890) are available online (https://dna.land/consent). To increase the database size, OpenSNP.org reference files can be downloaded. The DNA samples of NA12890 can be purchased – and DNA from YE001, SZ001 and JP001 can be requested, if desired, upon reasonable request.

For cell line authentication we refer to the CCLE database (1,099 reference files). Our method can be tested, and our data can be reproduced using that database and DNA from any cell line in the CCLE database.

Forensics databases are indeed more challenging to build compared to a cell line database. It would require a change in the methodologies currently employed, which might take years to actually happen. However, specific applications might be applicable more immediately. For instance, for sudden large-scale events (mass disasters, such as 9/11) SNP arrays could be employed, which is an efficient way to determine the variants of a single individual. This could be used in conjunction with MinION sketching for rapid identification of victims. We acknowledge that this will take a while to be implemented.

The first implementation opportunity of the method will be for cell line and tissue sample authentication – our method can be used in labs right away.

The second example given – that of cell line identification – does seem like a potential practical use, although at present the cost of such analysis would likely hinder its adoption compared with a simpler STR panel based approach for identification. Although the authors state that this is done by sending samples off and at great cost, this is also available to researchers to run in their own labs at not great cost (GenePrint system from ProMega at <$10/rxn). The authors may want to discuss how the cost could be brought down to similar levels.

The balance between costs of labor and costs of machine depreciation and consumables poses a trade-off for all methodologies. Yet, the requirement of extensive hands-on laboratory work seems to be a main driver for avoiding authentication tests. This, in turn, costs much more money down the line for the community as a whole because more irreproducible research is being published that is based on contaminated or mixed-up biological materials.

The Geneprint System requires hands-on work in the lab to amplify and check STR fragments. As described at length in multiple references we cite in the manuscript, somehow current authentication methods are not employed as routine laboratory tests. Moreover, the GenePrint system might be cheap on a per-sample basis, it does require the procurement of a capillary electrophoresis device from Thermo Fisher Applied Biosystems® that costs ~$120K. This stands in stark contrast to the MinION where the start-up costs are only $1000, after which consumables can be purchased (~$100 per sample) and used for the authentication method we present that requires only minimal costs of effort in the laboratory.

For the only relatively low number of samples that many laboratories need to authenticate the MinION will thus be more cost effective. By comparison, the method that makes use of the GenePrint and ThermoFisher Biosystem does not become cost efficient until more 1000 samples need to be authenticated.

The method as shown is also not likely to work well with lower levels of contamination and/or from contamination from multiple cell lines and does not seem to provide identification in such mixtures.

Many thanks for pointing this out. We added a paragraph in the Discussion:

“The main cause of cell line mix-ups is suggested to be human error (Alston-Roberts et al., 2010; Yu et al., 2015; Almeida et al., 2016). [...] The key to detect cell line contamination with human and non-human cells is periodic testing. “

In summary, I think to improve this article the authors should really spend time outlining the potential practical uses of this technique (including outside of human genetics) and then discuss in more detail the ethical concerns associated with such uses.

We thank the reviewer for elaborating on these points. We modified the manuscript and hope it conveys our message more clearly and convincingly.

First, we redirected the focus of our manuscript to an immediately implementable application of our method, namely cell line authentication. This will hopefully contribute to a major stride forward in reducing the use of unknown cell lines.

Second, for the forensics field:

We agree that the ethical issues are of crucial importance and highly relevant for the forensic applications and beyond. We describe these points in a previous version of our manuscript, which can still be found on BioRxiv:

http://www.biorxiv.org/content/early/2016/06/30/061556.article-metrics

We feel that this is outside the scope of the current version of our paper, which mostly focuses on cell line and tissue sample authentication and on certain applications in forensics.

The application of the method for other organisms is very interesting, and we are currently working on this. However, as we explained in our response to comments by reviewer 1, this would take too much away from our focus on cell line and tissue sample authentication, and we think such applications merit description in a separate manuscript.

[Editors' note: the author responses to the re-review follows.]

To be acceptable for publication, the reviewers have requested the following changes be made:1) Remove "and forensics" from the title.2) Remove "or in some forensic applications" from the title.3) Remove from the Introduction, the fourth paragraph describing forensics.4) Remove the paragraphs from the Discussion that discuss the utility of the approach for forensics, i.e. the first two paragraphs of the subsection “Forensics”.5) Remove "and to provide an alternative method for DNA-based forensics" from the concluding remarks.

We improved the manuscript as follows: We changed the title to: "Rapid Re-Identification of Human Samples Using Portable DNA Sequencing" and removed all sections that refer to forensics applications, as are listed by points 1-5.

In addition, please address the following issues:1) The reference database (http://files.teamerlich.org/pidp/CCLE_genotypes.tar.gz) should be deposited in a public repository (not the lab website).2) Concern has been raised over the availability of the code, which should be made publicly available by releasing it with some kind of Creative Commons license, for example.3) The competing interest statement (Y.E. is a consultant for a DNA forensic company) should be more explicit.

Thanks for pointing out we cannot publish a CCLE database via a lab website – we removed it, as it turns out we cannot re-publish the CCLE database. We provide the code that brings the reader immediately to the CCLE website where people can download the data.

We changed the GitHub code to the GPLv3 license.

We added specifics about Yaniv's affiliation to ArcBio.